# Matcha Green Tea Exhibits Bactericidal Activity against *Streptococcus pneumoniae* and Inhibits Functional Pneumolysin

**DOI:** 10.3390/antibiotics10121550

**Published:** 2021-12-17

**Authors:** Karin Sasagawa, Hisanori Domon, Rina Sakagami, Satoru Hirayama, Tomoki Maekawa, Toshihito Isono, Takumi Hiyoshi, Hikaru Tamura, Fumio Takizawa, Yoichi Fukushima, Koichi Tabeta, Yutaka Terao

**Affiliations:** 1Division of Microbiology and Infectious Diseases, Niigata University Graduate School of Medical and Dental Science, Niigata 951-8514, Japan; k-sasagawa@dent.niigata-u.ac.jp (K.S.); hisa-domon@dent.niigata-u.ac.jp (H.D.); d18a004k@mail.cc.niigata-u.ac.jp (R.S.); shirayama@dent.niigata-u.ac.jp (S.H.); maekawa-t@dent.niigata-u.ac.jp (T.M.); tisono@dent.niigata-u.ac.jp (T.I.); hiyoshi@dent.niigata-u.ac.jp (T.H.); h-tamura@dent.niigata-u.ac.jp (H.T.); ftakizawa@dent.niigata-u.ac.jp (F.T.); 2Division of Periodontology, Niigata University Graduate School of Medical and Dental Science, Niigata 951-8514, Japan; koichi@dent.niigata-u.ac.jp; 3Center for Advanced Oral Science, Niigata University Graduate School of Medical and Dental Sciences, Niigata 951-8514, Japan; 4Nestlé Japan Ltd., Wellness Communications, Tokyo 140-0002, Japan; yoichi.fukushima@jp.nestle.com

**Keywords:** *Streptococcus pneumoniae*, antibacterial resistance, pneumolysin, matcha green tea, catechin, epigallocatechin gallate, epigallocatechin, epicatechin gallate, epicatechin

## Abstract

*Streptococcus pneumoniae* is a causative pathogen of several human infectious diseases including community-acquired pneumonia. Pneumolysin (PLY), a pore-forming toxin, plays an important role in the pathogenesis of pneumococcal pneumonia. In recent years, the use of traditional natural substances for prevention has drawn attention because of the increasing antibacterial drug resistance of *S. pneumoniae*. According to some studies, green tea exhibits antibacterial and antitoxin activities. The polyphenols, namely the catechins epigallocatechin gallate (EGCG), epigallocatechin (EGC), epicatechin gallate (ECG), and epicatechin (EC) are largely responsible for these activities. Although matcha green tea provides more polyphenols than green tea infusions, its relationship with pneumococcal pneumonia remains unclear. In this study, we found that treatment with 20 mg/mL matcha supernatant exhibited significant antibacterial activity against *S. pneumoniae* regardless of antimicrobial resistance. In addition, the matcha supernatant suppressed PLY-mediated hemolysis and cytolysis by inhibiting PLY oligomerization. Moreover, the matcha supernatant and catechins inhibited PLY-mediated neutrophil death and the release of neutrophil elastase. These findings suggest that matcha green tea reduces the virulence of *S. pneumoniae* in vitro and may be a promising agent for the treatment of pneumococcal infections.

## 1. Introduction

*Streptococcus pneumoniae* is an important causative agent of invasive diseases and respiratory tract infections such as meningitis, sepsis, and pneumonia, and risk groups include young children, the elderly, and immunocompromised patients [1]. Antibiotics have been the first-choice method against pneumococcal infections; however, antimicrobial resistance is becoming more prevalent [2]. Our previous study showed that more than 80% of *S. pneumoniae* clinical isolates were non-susceptible to macrolides [3]. Therefore, it is desirable to develop new therapeutic approaches for the treatment of pneumococcal infections that do not rely on existing antimicrobial agents.

Pneumolysin (PLY), a cholesterol-dependent cytolysin, is a virulence factor of *S. pneumoniae* that is released from cells via autolysis [4,5]. PLY is found in virtually all pneumococcal isolates and plays a key role in bacterial colonization, invasion, and inflammation [6,7]. The functional PLY forms an oligomer, which then forms the transmembrane pores through the cholesterol-containing membranes, which lyse host cells and disrupt cellular functions [8,9]. When PLY binds to the membrane, it oligomerizes into a ring-shaped structure, the pre-pore complex, followed by the generation of oligomerized transmembrane pores [7,10]. PLY often shows strong cytotoxicity on red blood cells and neutrophils, thereby inducing disruption of pulmonary immune defenses and promoting pneumococcal infection [11]. A previous study has shown that the presence of anti-PLY antibodies provides a protective effect, delaying the time to first pneumococcal carriage in newborns [12]. These findings indicate that PLY is an important target for novel therapeutic agents against pneumococcal infections.

Novel treatments for pneumococcal infections that target PLY are under investigation. Previous studies have shown that the sub-minimum inhibitory concentrations (MICs) of macrolides reduce the release of PLY in highly macrolide-resistant *S. pneumoniae* [13,14,15]. Commonly prescribed cholesterol-lowering statins have been shown to reduce lung tissue damage by inhibiting PLY activity [16]. In addition, some natural compounds that inhibit PLY oligomerization have been reported to attenuate pneumococcal infection [17,18,19].

Tea is one of the most widely consumed beverages in many parts of the world. Its distinctive health-promoting effects are highly valued worldwide, as are its social-cultural connotations [20,21]. The health benefits of green tea for a wide variety of ailments such as anti-inflammatory and antibacterial effects have been investigated. Green tea consumption was reported to be associated with a lower mortality risk from pneumonia in Japanese women [22]. Gargling with green tea has also been shown to have protective effects against upper respiratory tract infections, and its potential efficacy needs to be investigated [23]. Matcha green tea is a specific form of Japanese green tea (*Camellia sinensis*) consumed as a thick suspension of a whole powdered green tea called “Tencha,” which is green tea leaves shaded for several weeks before harvest and dried without kneading [24]. This results in more antioxidant compounds and amino acids per cup consumption than “Sencha” green tea infusion, a more popular form of consumption [25,26,27]. It has also been reported that matcha green is rich in EGCG, one of the most active tea catechins [28]. Matcha green tea is expected to have a wide range of health benefits due to its characteristic composition of bioactive compounds. Additionally, tea catechins, a major component of green tea, show antibacterial activity against several bacteria such as *Salmonella*, *Clostridium*, and *Bacillus* [29]. However, the potential effect of matcha green tea on *S. pneumoniae* remains largely unknown.

In this study, we investigated the antibacterial activity of matcha green tea against antibiotic-susceptible and antibiotic-resistant *S. pneumoniae* strains. We also analyzed the inhibitory effect of matcha green tea on the cytotoxicity of PLY.

## 2. Results

### 2.1. Matcha Supernatants Exhibit Antibacterial Activity against Streptococcus pneumoniae

We first investigated the effect of matcha supernatant on the growth of *S. pneumoniae*. The recommended concentration for drinking matcha is 1.5 g of matcha powder with 70 mL of hot water [30], and therefore we used this concentration (20 mg/mL) as the standard for our in vitro analysis. Figure 1A shows that matcha supernatants inhibited the growth of both antibiotic-susceptible and -resistant pneumococcal strains in a dose-dependent manner. In particular, 20 mg/mL of matcha supernatant almost completely inhibited the growth of both antibiotic-susceptible and -resistant pneumococcus. Next, we examined whether the matcha supernatant showed bactericidal activity against *S. pneumoniae* strains. Figure 1B shows that 20 mg/mL of the matcha supernatant decreased the number of viable pneumococcal cells by more than 90%. Our findings indicate that the matcha supernatant at a concentration for regular consumption has direct bactericidal activity against *S. pneumoniae*, regardless of antibiotic resistance.

### 2.2. Matcha Supernatants Decrease Heolytic Activity of Recombinant PLY

We then analyzed the inhibitory effect of the matcha supernatant on the hemolytic activity of recombinant PLY. A series of experiments were performed by diluting the concentration of matcha supernatants to determine the minimum concentration of matcha supernatant that inhibited the hemolytic activity of rPLY. Figure 2A shows that the hemolytic activity of rPLY was significantly decreased by pre-incubation with matcha supernatants in a dose-dependent manner. The minimum concentration of the matcha supernatant that inhibited the hemolytic activity of rPLY was 40 μg/mL. Additionally, *S. pneumoniae* strain D39 was grown in the presence or absence of the 2.5 mg/mL of matcha supernatant, which did not affect the viability of the bacteria, until they reached a stationary phase, and the hemolytic activity of the bacterial supernatant was examined. Figure A1 shows that the hemolytic activity of the supernatant was significantly suppressed by treatment with the matcha supernatant. The addition of the matcha supernatant into bacterial supernatants from the untreated group also suppressed the hemolytic activity. Western blot analysis showed that the protein band representing the rPLY monomer was narrower and thinner depending on the concentration of the matcha supernatant (Figure 2B). In addition, the shading of the protein band of the rPLY monomer was different when comparing the group with 20 μg/mL of matcha supernatant and the group with more than 40 μg/mL of matcha supernatant. These results indicate that the matcha supernatant inhibited the hemolytic activity of both recombinant and native PLY. This also indicates that matcha supernatants may inhibit the detection of protein bands by the PLY antibody. Figure 2C shows that the sample containing only rPLY and the matcha supernatant contained the monomeric form. Moreover, it was shown that as the concentration of the matcha supernatant increased, the band of the monomer became narrower. As shown in the Coomassie brilliant blue-stained gel, there was no obvious difference in the protein bands between the concentrations of matcha supernatants (Figure 2D). These findings indicate that matcha supernatants inhibit the detection of protein bands by the anti-PLY antibody.

### 2.3. Matcha Supernatants Do Not Exhibit Cytotoxicity toward A549 Cells up to 5 mg/mL

We also analyzed the cytotoxicity of the matcha supernatant toward the human alveolar epithelial cell line, A549 (Appendix A). Figure A2 shows that the matcha supernatant did not show cytotoxicity toward A549 cells regardless of its concentration at 2 h. At 16 h, although ≥10 mg/mL of matcha supernatant significantly decreased A549 cell viability, 5 mg/mL of matcha supernatant did not affect the viability (Appendix B).

### 2.4. Matcha Supernatants Inhibit rPLY-Induced Neutrophil Death

Our previous study demonstrated that PLY induced lysis of human neutrophils [11]. Therefore, we investigated the protective effects of matcha supernatants on PLY-induced neutrophil death. Figure 3 shows that treatment with rPLY significantly induced neutrophil death. Although rPLY induced cell injury and death, co-incubation with 1 and 5 mg/mL of the matcha supernatant significantly increased the number of viable cells (Figure 3A,B). These findings indicate that the matcha supernatant significantly inhibits PLY-mediated neutrophil injury.

### 2.5. Catechins Suppress the Hemolytic Activity of rPLY by Inhibiting Its Oligomerization

A previous study showed that epigallocatechin gallate (EGCG) in matcha green tea effectively inhibits the hemolytic activity of PLY and that a novel strategy that targets PLY using EGCG is a promising therapeutic option for *S. pneumoniae* [31]. We hypothesized that catechins, the major components of matcha green tea, are involved in the inhibition of the cytotoxicity of rPLY. As shown in Table 1, matcha green tea contains various catechins. Additionally, a cup of matcha green tea contained a higher amount of EGCG and ECG than that of sencha green tea. We selected EGCG, epigallocatechin (EGC), epicatechin gallate (ECG), and epicatechin (EC), which are abundant catechins in matcha green tea, and examined their effects on the hemolytic activity of rPLY. Figure 4 shows that EGCG, EGC, and ECG significantly inhibited the hemolytic activity of rPLY in a concentration-dependent manner, while EC had no effect. EGCG significantly inhibited the hemolytic activity of rPLY at concentrations above 0.13 μg/mL, which is within the concentration of 2.17 μg/mL of matcha supernatant. Similarly, EGC and ECG affected the hemolytic activity of rPLY. It has been reported that galloylated catechins such as EGCG and ECG are strong inhibitors of protein activity, and catechins with catechol groups such as EC and ECG have lower antioxidant potential compared to those with pyrogallol groups such as EGCG and EGC [32,33,34]. Our results are consistent with these data and indicate that catechins are involved in the inhibition of rPLY hemolytic activity by matcha supernatant and that the strength of the inhibition varies depending on the type of catechin. Oligomerization analysis was performed to determine whether catechins affect the pre-pore complex of rPLY and mitigate its biological toxicity. Figure 5A–C shows that both oligomeric and monomeric bands of rPLY became thinner after EGCG, EGC, and ECG treatment. Meanwhile, there was no significant change in the oligomeric and monomeric bands when the concentration of EC was increased (Figure 5D). Consequently, EGCG, EGC, and ECG suppress the hemolytic activity of rPLY by inhibiting its oligomerization.

### 2.6. Catechins Suppress rPLY-Induced Loss of Neutrophil Viability

Next, we analyzed the inhibitory effect of catechins on rPLY-induced neutrophil death. As shown in Figure 6A–D, treatment of neutrophils with EGCG, EGC, and ECG markedly inhibited rPLY-induced neutrophil death compared to the untreated group. The number of viable neutrophils tended to increase as the concentration of added catechins increased. In contrast, the degree of increase in the number of viable neutrophils with EC treatment was moderate compared to that of EGCG, EGC, and ECG (Figure 6E). These findings suggest that catechins have an inhibitory effect on the cytotoxicity of rPLY.

### 2.7. Catechins Suppress the Release of Neutrophil Elastase in Response to rPLY Stimulation

We previously reported that PLY-induced neutrophil death leads to the release of neutrophil elastase (NE), which subsequently damages the surrounding tissues and causes lung dysfunction associated with pneumonia [35]. Based on Figure 6, we hypothesized that catechins might also inhibit the release of NE by rPLY and performed an NE activity assay in the neutrophil-culture supernatant to test this. Figure 7 shows that all catechins significantly suppressed NE activity in the supernatants obtained from catechin-treated human neutrophils compared to those obtained from neutrophils treated with rPLY alone.

## 3. Discussion

In the present study, the matcha supernatant showed antibacterial activity against the *S. pneumoniae* strains examined. In addition, the matcha supernatant and catechins inhibited the oligomerization of PLY and suppressed its cytotoxic effect by inhibiting neutrophil death, and the release of NE was suppressed. Furthermore, catechins in matcha green tea inhibited PLY oligomerization and reduced the virulence of PLY in vitro. Thus, our data showed the possibility that components of matcha green tea exhibit antibacterial and antitoxin activities against *S. pneumoniae*.

A variety of studies have been conducted to determine the antibacterial activity of green tea, and the results of these studies show that green tea might be effective against many Gram-positive and Gram-negative organisms as well as a few viruses, fungi, and parasites [36,37]. Green tea exhibits antimicrobial activity against clinical isolates of multidrug-resistant *Staphylococcus aureus* and *Escherichia coli*, suggesting its potential use as an antimicrobial agent [38,39]. The beneficial components of green tea have been characterized and identified to be catechins. The antibacterial activity of catechins is attributed to their binding to the cell membrane of the bacterial lipid bilayer and damage to the membrane [40,41]. EGCG is found at high concentrations in green tea and is the main active catechin [20]. As shown in Table 1, matcha powder contains various catechins, most of which are EGCG. In this study, the first approach was to demonstrate whether the matcha supernatant had a bactericidal effect on *S. pneumoniae* strains and inhibitory effect on bacterial growth. It was found that the matcha supernatant inhibited bacterial growth of antibiotic-resistant and -susceptible S. pneumoniae strains in a dose-dependent manner. In addition, the matcha supernatant showed bactericidal activity against *S. pneumoniae* strains, and we conclude that the inhibition of bacterial growth by the matcha supernatant was mainly due to the bactericidal activity. It is also possible that the susceptibility of *S. pneumoniae* strains D39 and KM256 to the antibacterial activity of the matcha supernatant is different. To our knowledge, this study is the first to report the bactericidal effect of matcha green tea against multidrug-resistant pneumococci. Our in vitro findings also indicated that the matcha supernatant did not exhibit cytotoxicity against human alveolar epithelial cells at 2 h. However, prolonged exposure for 16 h to the matcha supernatant showed cytotoxicity. In the present study, it is difficult to elucidate the detailed mechanism of the bactericidal effect of the matcha supernatant, and further research on the effect of matcha green tea on pneumococcal infection is needed.

Previous studies have indicated that PLY is required for the pathogenesis of pneumonia caused by pneumococcal infections. PLY can bind to cholesterol on cell membranes and oligomerize to form pore complexes in the cytomembrane, a critical process that can cause cytolysis [7]. It has been suggested that PLY-deficient *S. pneumoniae* strains do not cause lung tissue damage or promote bacterial growth in vivo [42]. These findings demonstrated the efficacy of PLY in inhibiting the cytotoxicity of pneumococcal infections, and the study of PLY as a therapeutic target is expected to address pneumococcal infections. Our findings indicate that the matcha supernatant inhibits the hemolytic activity of PLY by inhibiting PLY oligomerization. The anti-PLY antibody clone PLY-4 recognizes the epitope involving Arg 232, which is a part of an exposed loop on the edge of the concave and convex faces of domain 1 [43]. Domain 1 is thought to be involved in PLY oligomerization [44]. Our findings also showed that the monomeric bands of PLY became thinner depending on the concentration of the matcha supernatant. These results suggest that the components of the matcha supernatant competitively antagonize PLY-4 and that these components suppress the hemolytic activity of PLY.

Catechins in matcha green tea are known to exhibit various beneficial health properties. EGCG, EGC, ECG, and EC are the main bioactive catechin components of matcha green tea [20,28,45]. A previous study has shown that EGCG has pharmacological effects such as anti-inflammatory and antibacterial effects, and can neutralize the hemolytic activity of PLY by inhibiting its oligomerization [31]. Consistent with this study, our findings also showed that EGCG neutralized the cytotoxicity of PLY by inhibiting its oligomerization. In addition, EGC and ECG showed inhibitory effects on the hemolytic activity of PLY, diluting not only the oligomeric bands but also the monomeric bands in western blotting. These findings suggest that EGC and ECG have competitive antagonistic effects on PLY-4. In contrast, EC did not inhibit the hemolytic activity of PLY by inhibiting its oligomerization, suggesting that the inhibitory activity against PLY differs among these catechins.

NE has been reported to play an important role in killing intracellular bacteria by neutrophils; however, excessive release of NE damages the surrounding tissues, leading to lung dysfunction associated with bacterial pneumonia. Our previous study demonstrated that neutrophils were more sensitive to PLY-induced cell lysis than A549, alveolar epithelial cells [11]. NE is eventually released from dead neutrophils [46,47], resulting in the cleavage of multiple host proteins [35]. Several studies have shown that NE inhibition reduced lung injury in animal models [48]. According to our data, matcha supernatant and catechins (EGCG, EGC, ECG and EC) can enhance the survival rate of neutrophils and reduce the release of NE from neutrophils by neutralizing the cytotoxicity of PLY in vitro. EGCG administration was reported to provide protection against pneumococcal pneumonia in mice and reduce the pathological injury and bacterial burden in the lungs [31]. Recent studies have suggested that EGEG directly binds to NE and inhibits its enzymatic activity in a concentration-dependent manner [49,50]. To the best of our knowledge, this study is the first to show that EGC and ECG suppress the cytotoxicity of PLY and inhibit the NE release. Taken together, our findings indicate that the inhibition of PLY cytotoxicity on neutrophils by matcha green tea catechins may lead to a reduction in the virulence of pneumococci. Further studies are required to elucidate the mechanisms by which catechins inhibit the cytotoxicity of PLY.

Our study has several limitations. We removed the tea leaf powder from the matcha samples and used the supernatant only considering that the presence of the powder could interfere with the measurement of bacterial turbidity. Additionally, the temperature and time for the extraction of the matcha supernatant differ from those recommended for drinking. Therefore, our study might not be directly translated to a person taking it.

In conclusion, our findings demonstrate that the concentration of 20 mg/mL matcha green tea has bactericidal effects on *S. pneumoniae* regardless of its antimicrobial resistance and has inhibitory effects on the pore-forming toxin PLY in vitro. Studies on the antimicrobial effects of matcha may provide promising data for novel strategies to prevent or improve the therapeutic outcome for *S. pneumoniae*, which is becoming increasingly drug-resistant. The sequences of the *ply* gene and its products are highly conserved across pneumococci [51]; therefore, it is suggested that matcha green tea may reduce the cytotoxicity of PLY regardless of the pneumococcal strain. Further studies are needed to investigate the effects of matcha green tea and its specific components on antimicrobial-resistant pneumococcal pneumonia.

## 4. Materials and Methods

### 4.1. Bacterial Strains and Reagents

Multidrug-resistant *S. pneumoniae* strain KM256 (penicillin G, ceftriaxone, azithromycin, and levofloxacin MICs of >8, 8, 4, and >8 μg/mL, respectively), which was isolated from the nasopharynx of patients with acute otitis media [3], and antibiotic-susceptible *S. pneumoniae* strain D39, were grown in tryptic soy broth (TSB) at 37 °C for 12 h without shaking [52]. The overnight cultures were then inoculated into fresh TSB and allowed to grow until they reached the exponential growth phase (optical density at 600 nm of 0.1). The bacteria were subsequently used for antimicrobial and bactericidal activity assays. Matcha green tea powder was kindly provided by Kyoeiseicha Co. Ltd. (Kyoto, Japan). The powder was suspended in TSB or phosphate-buffered saline (PBS) and boiled at 65 °C for 120 min. The matcha supernatants were collected by additional centrifugation at 5000× *g* for 10 min. In our preliminary experiments, the matcha supernatant prepared by heating at 65 °C for 120 min showed the highest bactericidal effect on *S. pneumoniae* strain D39. Therefore, we decided to prepare the matcha supernatant by heating at 65 °C for 120 min and using it for the following analysis. EGCG, EGC, ECG, and EC were purchased from Nagara Science Co. Ltd. (Gifu, Japan) and dissolved in PBS. rPLY was prepared as previously described [11].

### 4.2. Effect of Matcha Supernatant on the Growth of S. pneumoniae

*S. pneumoniae* strains were grown until the log phase (OD600 = 0.1) and inoculated into 5 mL of TSB. These bacterial cultures were then treated with TSB (as the untreated) or 1.25–20 mg/mL of matcha supernatants and incubated at 37 °C. At each time point, bacterial growth was measured at a wavelength of 600 nm using a Mini Photo 518R (Taitec, Tokyo, Japan).

### 4.3. Hemolytic Assay

Fresh sheep erythrocytes were centrifuged at 450× *g* for 10 min and washed three times with PBS. Erythrocytes (1% *v*/*v*) in PBS were mixed with rPLY (1 μg/mL) in the presence or absence of varying concentrations of matcha supernatant (20–300 μg/mL) or catechins (0.1, or 1 mg/mL). The samples were incubated at 37 °C for 30 min and centrifuged at 450× *g* for 10 min. The supernatant was pipetted into flat-bottomed microtiter plates, and hemolysis was measured using a microplate reader (Thermo Fisher Scientific, Waltham, MA, USA). A hemolytic unit is defined as the amount of rPLY contained in 1 mL of PBS that causes 50% lysis of a 1% erythrocyte suspension after incubation at 37 °C for 30 min [53].

### 4.4. Human Neutrophil Isolation

Neutrophils were prepared as described previously [54]. Briefly, heparinized blood was obtained from healthy donors and layered onto Polymorphprep^TM^ (Axis Shield, Dundee, UK) at a 1:1 ratio. After centrifugation at 500× *g* for 30 min, the layer containing neutrophils was collected, and residual red blood cells were hypotonically lysed. Viable cells were monitored using the Trypan blue exclusion method, and the cells were counted using a Countess II automated cell counter (Thermo Fisher Scientific, Waltham, MA, USA). The experimental protocol was approved by the Institutional Review Board of Niigata University, and the methods were carried out in accordance with the approved guidelines (permit # 2018-0075). Informed consent was obtained from all donors prior to inclusion in this study.

### 4.5. Cytotoxic Assay

Human neutrophils (2.5 × 10^5^ cells/200 μL) were stimulated with rPLY (1 μg/mL) in the presence or absence of the varying concentrations of the matcha supernatant (20–320 μg/mL) or catechins (0.1, or 1 mg/mL) at 37 °C for 1 h. After incubation and treatment with the Apoptotic/Necrotic/Healthy Cells Detection Kit (PromoCell, Heidelberg, Germany), the samples were observed and photographed using a confocal laser scanning microscope (Carl Zeiss, Jena, Germany) to evaluate cell viability.

### 4.6. Western Blotting Assay

rPLY (1 µg/mL) and varying concentrations of matcha supernatant (20–320 μg/mL) or catechins (0.1, or 1 mg/mL) in PBS were mixed and incubated at 37 °C for 1 h. Then, sheep erythrocytes were diluted with the supernatant to a final concentration of 1% (*v*/*v*) and incubated at 37 °C for 1 h. Thereafter, the samples were suspended in a 4 × sodium dodecyl sulfate-polyacrylamide gel electrophoresis (SDS-PAGE) sample buffer (200 mM Tris-HCl [pH 6.8], 8% SDS, and 0.02% bromophenol blue) without β-mercaptoethanol and incubated at 50 °C for 10 min. After incubation, protein samples (20 μL each) were separated on a 7.5% SDS-PAGE gel (Bio-Rad Laboratories, Hercules, CA, USA) and transferred onto a polyvinylidene fluoride membrane (Merck Millipore, Billerica, MA, USA) at 80 V for 90 min. The membranes were then incubated with a blocking reagent (Nacalai Tesque, Kyoto, Japan) to block nonspecific binding, incubated with anti-PLY monoclonal antibody clone PLY-4 (1:1000; Abcam, Cambridge, UK), and probed with a horseradish peroxidase (HRP)-conjugated secondary anti-mouse antibody (1:3000; Cell Signaling Technology, Danvers, MA, USA). Thereafter, the membranes were treated with HRP substrates (GE Healthcare, Little Chalfont, UK) and imaged using an ImageQuant LAS 4000 (GE Healthcare Bio-Sciences AB, Uppsala, Sweden).

### 4.7. Protein Staining

Varying concentrations of matcha supernatant (200–3200 μg/mL) and rPLY (10 μg/mL) in PBS were mixed and incubated at 37 °C for 1 h. The samples were then suspended in a 4 × SDS-PAGE sample buffer (200 mM Tris-HCl, pH 6.8, 8% SDS, 0.02% bromophenol blue, and 4% β-mercaptoethanol) and heated at 95 °C for 5 min. The samples (20 μL each) were separated on 12% SDS-PAGE gel (Bio-Rad Laboratories) at a constant current of 20 mA per gel, and the gel was stained with a Coomassie Brilliant Blue Stain Kit (Integral, Tokushima, Japan).

### 4.8. Neutrophil Elastase Activity Assay

NE activity in the neutrophil culture supernatant was evaluated using the specific substrate N-methoxysuccinyl-Ala-Ala-Pro-Val p-nitroanilide (Merck Millipore) as described previously [11]. Briefly, the samples were incubated with 0.1 M Tris-HCl (pH 8.0) containing 0.5 M NaCl and 1 mM substrate at 37 °C for 6 h, and the absorbance was measured at 405 nm.

### 4.9. Statistical Analysis

Data were statistically analyzed via analysis of variance with Dunnett’s or Tukey’s multiple-comparison tests or Student’s *t*-test using GraphPad Prism software (version 8.4.3; GraphPad Software, Inc., La Jolla, CA, USA).

## Figures and Tables

**Figure 1 antibiotics-10-01550-f001:**
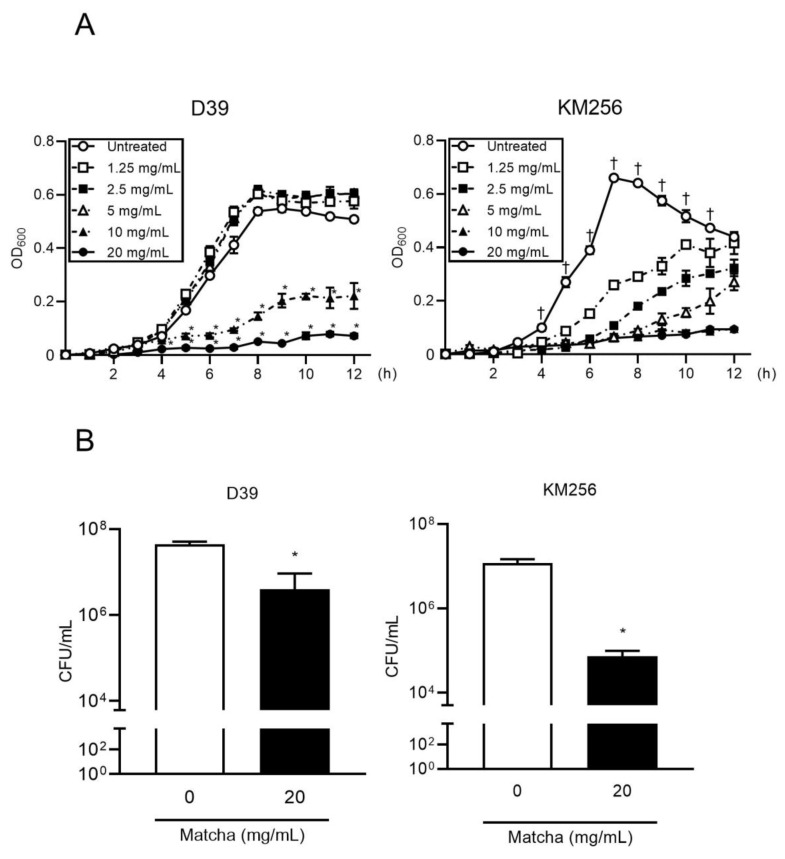
Matcha supernatants inhibit the growth of *Streptococcus pneumoniae*. (**A**) *S. pneumoniae* strains D39 and KM256 were cultured in the presence or absence of matcha supernatants (1.25–20 mg/mL) suspended in Trypticase Soy Broth (TSB). Optical density at 600 nm (OD_600_) of each sample was measured at each time point. Data are shown as the mean ± standard deviation (SD; *n* = 3 per group) and were evaluated using two-way analysis of variance with Tukey’s multiple-comparisons test. * *p* < 0.05 versus all other groups; † *p* < 0.05 versus all matcha supernatant-treated groups. (**B**) *S. pneumoniae* strains D39 and KM256 were grown in TSB medium until an OD_600_ of 0.1 was reached. Thereafter, the bacterial suspensions were incubated at 37 °C for 2 h in matcha supernatant (20 mg/mL). The treated bacteria were plated on sheep blood agar plates, and the colony-forming units were determined. Data are presented as the mean ± SD (*n* = 3) and assessed using the Student’s *t*-test. * *p* < 0.05 versus the untreated group.

**Figure 2 antibiotics-10-01550-f002:**
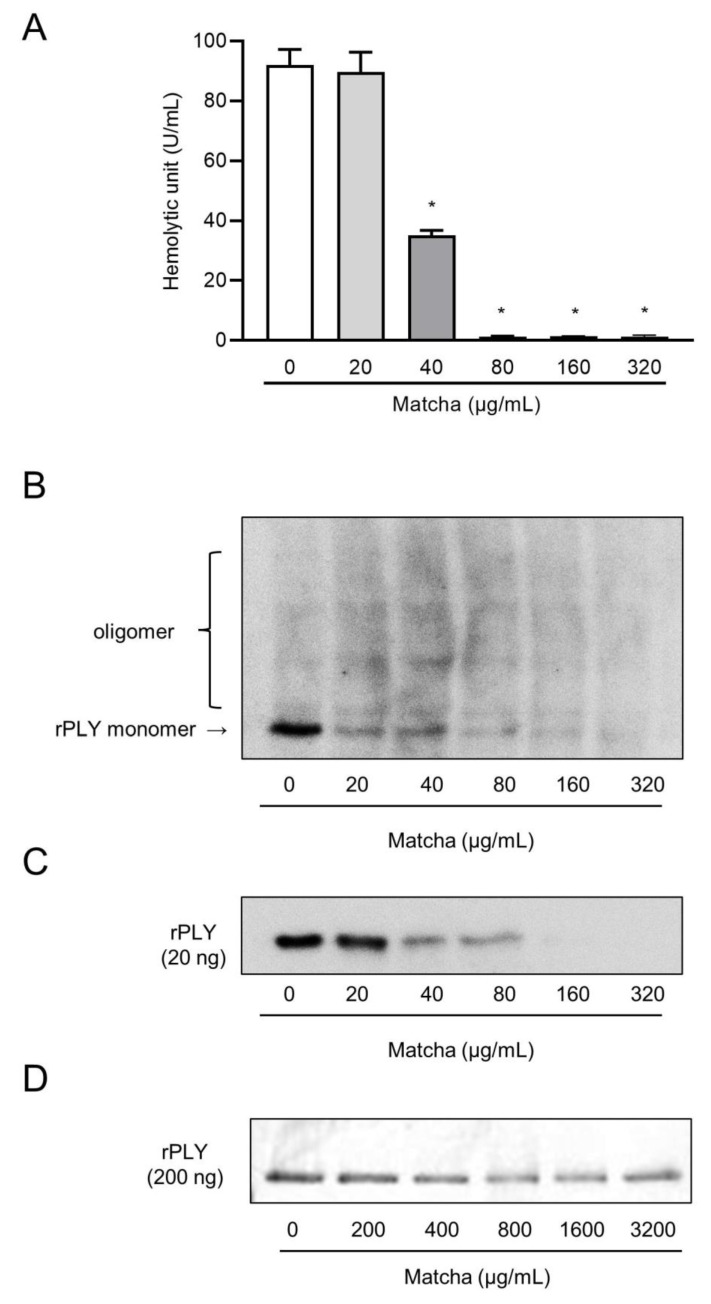
Matcha supernatants decrease PLY-induced hemolysis. Recombinant PLY (rPLY; 1 μg/mL) was preincubated with varying concentrations of matcha supernatant at 37 °C for 1 h. (**A**) The hemolytic activity in each sample was determined. Data are shown as the mean ± SD (*n* = 3 per group) and were evaluated using one-way analysis of Dunnett’s multiple-comparisons test; * *p* < 0.05 versus the untreated group. (**B**) PLY oligomerization of the samples was detected by western blotting using the anti-PLY antibody clone PLY-4. The experiment was repeated three times, and a representative result is shown. rPLY was co-incubated with the matcha supernatant at 37 °C for 1 h, and the PLY of the samples was detected (**C**) by western blotting and (**D**) Coomassie Brilliant Blue staining.

**Figure 3 antibiotics-10-01550-f003:**
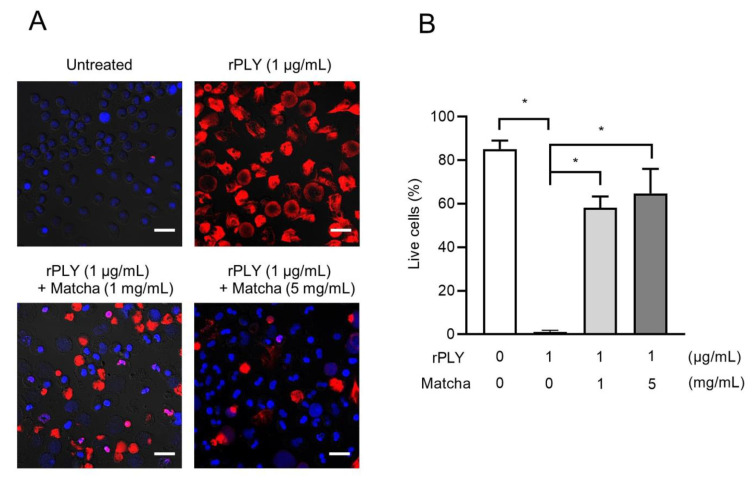
Matcha supernatants attenuate PLY-mediated neutrophil death. (**A**) Human neutrophils were treated with rPLY(1 μg/mL) in the presence or absence of matcha supernatants (1 and 5 mg/mL) at 37 °C for 1 h. Representative fluorescence images of cells stained by Hoechst 33342 (viable cells; blue) and ethidium homodimer III (necrotic cells; red) are shown. Scale bars: 20 μm. (**B**) The percentage of Hoechst 33342-positive cells was calculated. Data are presented as the mean ± SD of quintuplicate determinants and were evaluated using one-way analyses of variance with Dunnett’s multiple-comparison test; * *p* < 0.05 versus the PLY only-treated group.

**Figure 4 antibiotics-10-01550-f004:**
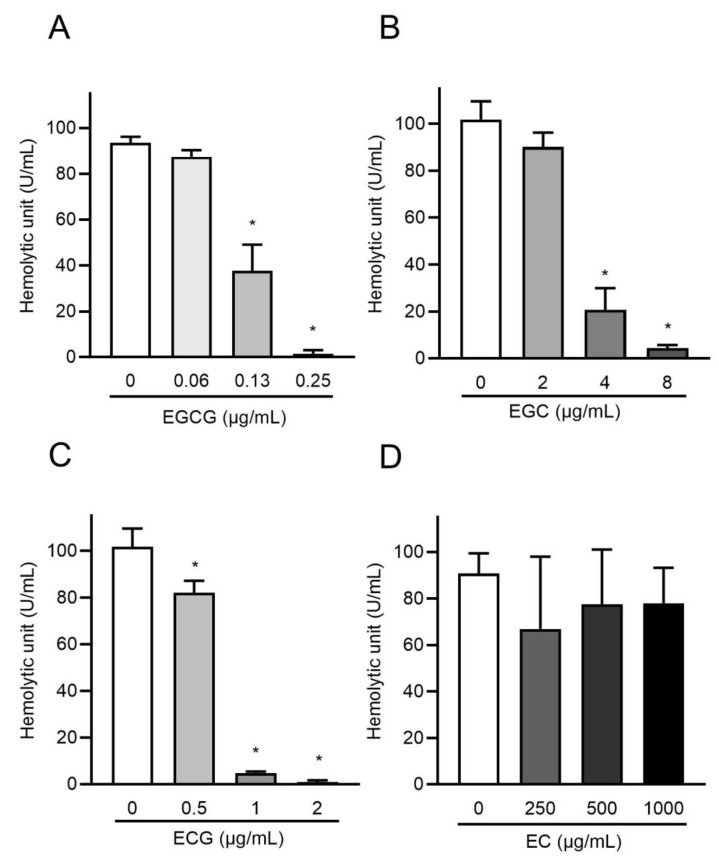
EGCG, EGC, and ECG decrease the hemolytic activity of PLY. rPLY (1 μg/mL) was preincubated with varying concentrations of (**A**) EGCG, (**B**) EGC, (**C**) ECG, and (**D**) EC at 37 °C for 1 h, after which the hemolytic activity of each sample was determined. Data are shown as the mean ± SD (*n* = 3 per group) and were evaluated using one-way analysis of Dunnett’s multiple-comparisons test, * *p* < 0.05 versus the untreated group.

**Figure 5 antibiotics-10-01550-f005:**
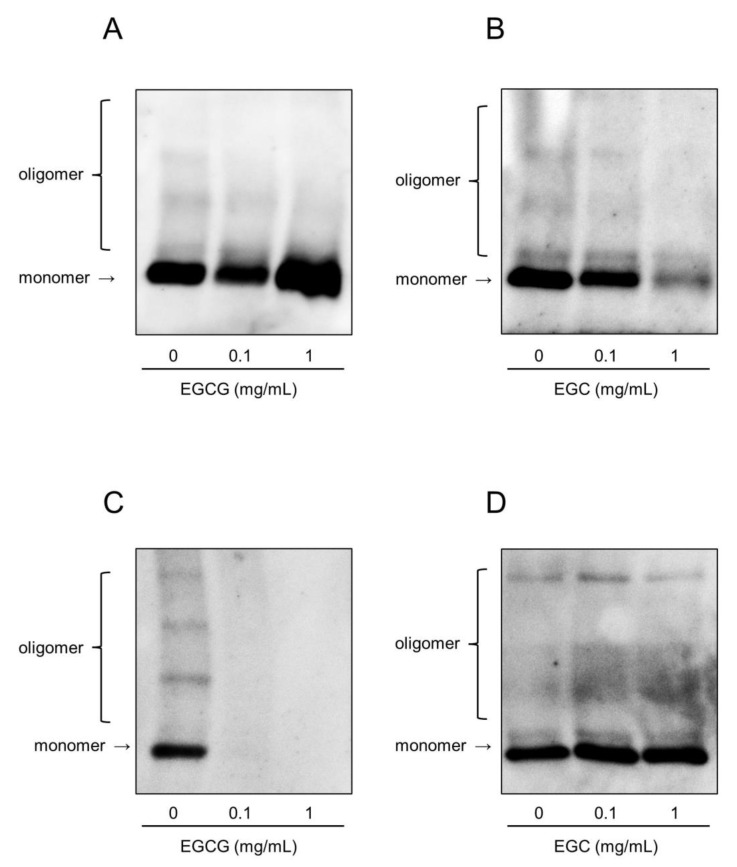
EGCG, EGC, and ECG inhibit PLY oligomerization. rPLY (1 μg/mL) was incubated with 1% (*v*/*v*) sheep erythrocytes and 0.1 or 1 mg/mL of (**A**) EGCG, (**B**) EGC, (**C**) ECG, and (**D**) EC, and PLY oligomerization was analyzed by western blotting. The experiment was repeated three times, and a representative result is shown.

**Figure 6 antibiotics-10-01550-f006:**
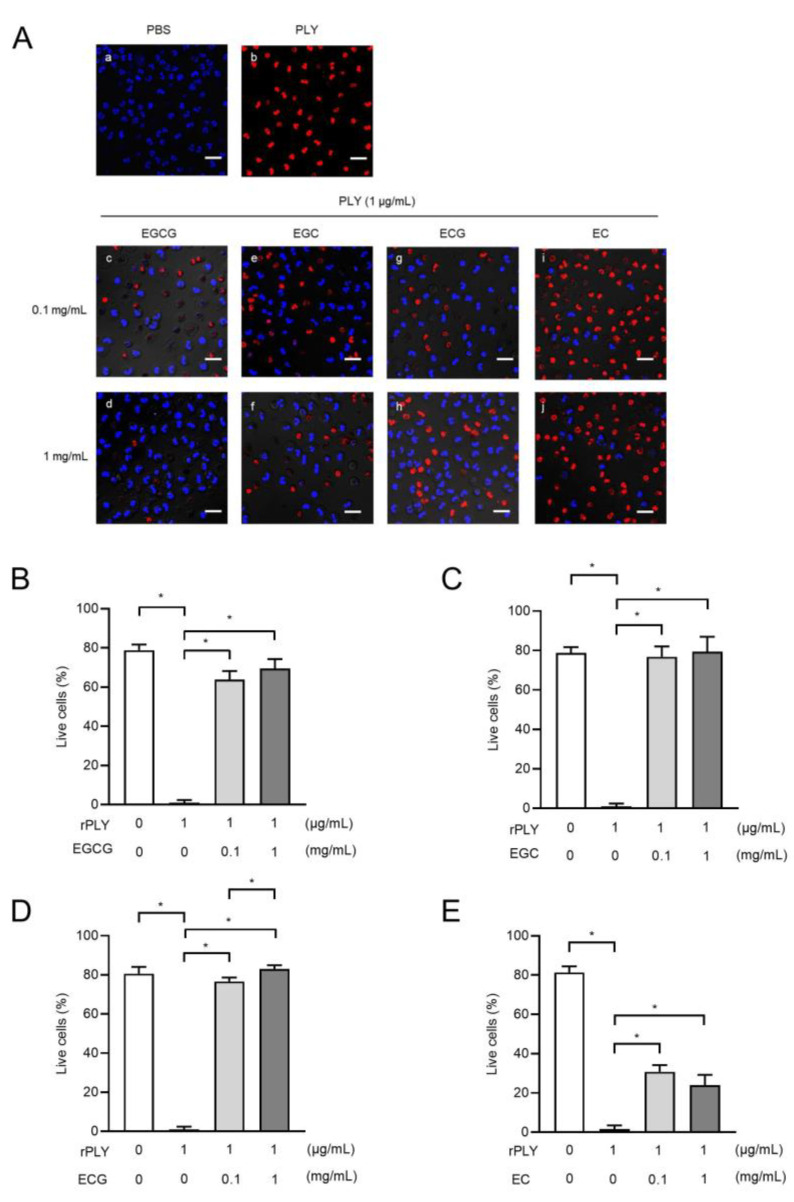
Catechins attenuate PLY-mediated human neutrophil death. (**A**) Human neutrophils were treated with rPLY (1 μg/mL) in the presence or absence of catechins (0.1 and 1 mg/mL) at 37 °C for 1 h. Representative fluorescence images of cells stained by Hoechst 33342 (viable cells; blue) and ethidium homodimer III (necrotic cells; red) are shown. Scale bars: 20 μm. (**B**–**E**) The percentage of Hoechst 33342-positive cells were calculated. Data are presented as the mean ± SD (*n* = 5 per group) and were evaluated using one-way analyses of variance with Dunnett’s multiple-comparison tests. * *p* < 0.05 versus the PLY-treated group.

**Figure 7 antibiotics-10-01550-f007:**
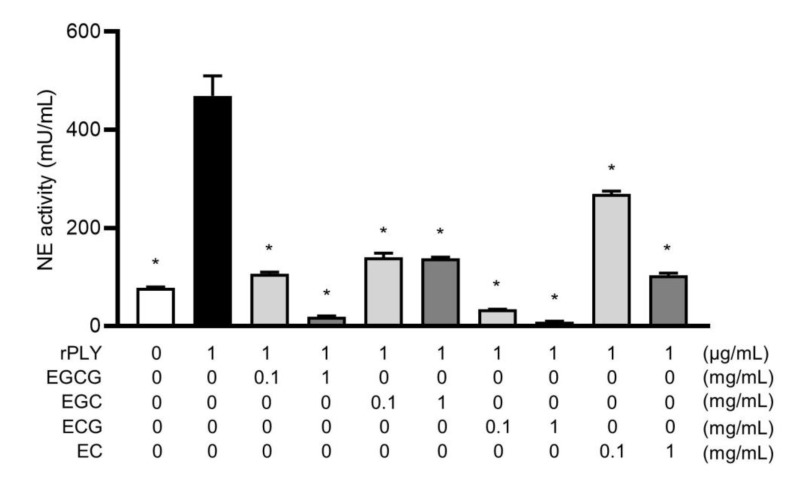
Catechins decreased recombinant PLY-stimulated NE release from neutrophils. Human neutrophils (5 × 10^5^ cells/200 μL) were exposed to rPLY (1 μg/mL) at 37 °C for 1 h in the presence or absence of EGCG, EGC, ECG, and EC (0.1 and 1 mg/mL). NE activity in the culture supernatant was evaluated using an elastase activity assay kit. Data are presented as the mean ± SD (*n* = 3 per group) and were evaluated using one-way analysis of Dunnett’s multiple-comparisons test. * *p* < 0.05 versus the PLY only-treated group.

**Table 1 antibiotics-10-01550-t001:** Composition of matcha green tea powder and dried Sencha green tea infusion per cup consumption.

Compounds	Unit	Matcha Green Tea Powder *	Sencha Green Tea Infusion, Dried **
		per 1.5 g	per 0.3 g
Energy	kcal	3.0	0.6
Protein	g	0.43	0.02
Fat	g	0.09	0.002
Ash	g	0.10	0.03
Carbohydrates	g	0.66	0.13
Dietary fiber	g	0.47	0.01
Caffeine	mg	43.5	14.7
Vitamin C	mg	2.85	0.63
Vitamin K	mg	0.046	0.0001
Carotenoids	mg	2.24	0.01
Lutein	mg	1.16	0.003
Theanine	mg	26.4	4.6
Chlorophylls	mg	14.3	0.006
Catechin (C)	mg	0.6	1.7
Epicatechin (EC)	mg	7.1	6.3
Gallocatechin (GC)	mg	1.7	9.3
Epigallocatechin (EGC)	mg	30.0	26.7
Epigallocatechin gallate (EGCG)	mg	88.5	20.4
Gallocatechin gallate (GCG)	mg	1.0	4.5
Epicatehin gallate (ECG)	mg	15.0	3.3
Catechin gallate (CG)	mg	0.1	0.5

* Matcha green tea, generally served at 1.5 g powder in 70 mL hot water (about 80 °C). ** Commercially available dried Sencha green tea infusion, NIKKEN FOOD (DG-SP100), served at 0.3 g per 100 mL.

## Data Availability

All data are contained within the manuscript.

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
