# Peer review of "Matcha Green Tea Exhibits Bactericidal Activity against Streptococcus pneumoniae and Inhibits Functional Pneumolysin"

_antibiotics, 2021, doi:10.3390/antibiotics10121550_

Round 1

Reviewer 1 Report

The current article by Sasagawa et al. examines the effects of matcha green tea has on Streptococcus pneumoniae. Specifically they examine bactericidal activity, inhibition of pneumolysin oligomerization and toxicity, as well as the effect of individual extracts of matcha tea on pneumolysin mediated hemolysis. Overall the paper is well written and they progression of experiments performed is logical. There are several strengths to the current article, but some issues should be addressed before being considered before publication. These are examined in more detail below.

Major:

Further discussion or speculation on the direct effects matcha tea and its extracts are having on pneumolysin are needed. Figure 2 and all other Western blot experiments tend to indicate reduction in not only oligomerization, but also total pneumolysin present. This trend is seen at increasing concentrations of the used inhibitor. Figure 2D using a coomassie stain shows that pneumolysin is present at similar levels upon incubation at high levels of matcha, but the size does not change which would indicate binding of matcha extracts to the protein. If the matcha extracts are binding the pneumolysin and inhibiting antibody recognition of specific sites during Western blot then these extracts should still be binding the rPLY in the coomassie stain and increase relative protein size. It is not clear what effect the matcha is having on rPLY and this needs to be addressed.

It is interesting that in Figure 1 the drug resistant strain KM256 is much more susceptible to matcha extract than the highly sensitive lab strain D39, which seems to grow better at low levels compared to untreated. A small discussion talking about strain variability and possible cause would be appreciated and add context to the findings.

More information in the introduction is necessary on matcha green tea and why it is useful to examine antimicrobial effects, more that saying it is rich in antioxidants. Some elaboration is giving in the results, but the importance of green tea is barely discussed in the introduction.

No experiment is done examining the effect of matcha extracts and cell viability using whole cells, only purified pneumolysin. A validating experiment with live pneumococcal cells and matcha extracts and determining viability is necessary. D39 viability is not effected at 2.5 mg/ml concertation and 1 mg/ml of matcha was able to significantly increase neutrophil survival upon rPLY addition

Minor:

Ln. 66. “methods” spelled incorrectly

Ln. 78. Add “bacteria” or “pneumococcus” after the phrase “both antibiotic-susceptible and –resistant.”

Figure 1A. The 5 mg/ml dosage on D39 strain does not seem to be present.

Ln. 251. “PLY” should not be capitalized.

Author Response

We thank the reviewer for their comments. Point-by-point responses to comments are detailed in the attached.

Reviewer 2 Report

This paper reported the bactericidal activity of matcha green tea against against S. pneumoniae and its link with pneumolysin. Although relevant, I believe this manuscript would benefit from some changes as per my suggestions below:

Line 31 "might be a promising compound" - Authors may want to refer to matcha as a whole and not a single compound.

Line 76 - Should be "in a dose-dependent manner"

Line 77 - Concentrations above 1 mg/mL are very high, even for full extracts and not purified compounds. How do the authors explain this value? Are there any comparative data? Is matcha better than green tea? Does Figure 1 have any control? Very high extract concentrations may reduce the bacterial growth but induce toxicity. A critical review on these numbers is needed in the discussion.

Line 80 - I am afraid this extrapolation needs some context. Not all drinking matcha is "available" against S. pneumoniae in a clinical context. ADME should be considered. Thus, this sentence may be misleading and should be deleted. Line 82/83 should be re-phased.

Line 139 - EGCG is also in green tea. Authors should consider referring some differences between matcha green tea and green tea here. Otherwise this subchapter 2.4. can be related to any tea (already studied, no novelty).

Line 155 - Is there any reference for table 1? Are these compounds (and percentages) present in the powder or in the solution after "suspended in TSB or phosphate-buffered saline (PBS) and boiled at 65°C for 120 min" (line 263).

Line 263 - This preparation of matcha green tea does not match with what is done prior consumption. Thus, my comment above regarding the mislead extrapolation gain even more relevance. The results here showed cannot be translated to what/how a person takes it. 

A quick note on why the preparation of this tea is done with TSB/PBS and boiled at 65C for 120min should be done.

Line 248 and Abstract - A focus on the high dose required for the antibacterial effects should be given.

Line 253 - I would not agree with "matcha green tea might be a (...) strategy". The activity of specific component of matcha green tea against antimicrobial resistant S. pneumoniae should be more researched. Authors should also focus on the differences between matcha green tea and green tea of other teas. 

Are there any references for antibacterial activity of tea components against resistant S. pneumoniae?

Author Response

(The authors gave the same response as above.)

Round 2

Reviewer 2 Report

The authors responded very well to all the suggestions. Very well done. I have no issues at all with the reviewed manuscript.